## Rapid Communication

women; war; mental health; Ukraine; Israel

**Corresponding author:**
Richard Isralowitz;
Email: Richard@bgu.ac.il

# War impact on mental health and well-being among Ukrainian and Israeli women: A cross-national comparison

Alexander Reznik[1], Valentyna Pavlenko[2], Anton Kurapov[3], Liudmyla Zavatska[4], Nataliia Korchakova[5], Iuliia Pavlova[6] , Shai-li Romem-Porat[7] and Richard Isralowitz[7]

[1]Regional Alcohol and Drug Abuse Research (RADAR) Center, Ben-Gurion University of the Negev, Be'er Sheva, Israel; [2]Department of Applied Psychology, V.N. Karazin Kharkiv National University, Kharkiv, Ukraine; [3]Department of Experimental and Applied Psychology, Taras Shevchenko National University of Kyiv, Kyiv, Ukraine; [4]Department of Social Work, Educational and Pedagogical Sciences, T.H. Shevchenko National University "Chernihiv Colehium", Chernihiv, Ukraine; [5]Department of Developmental & Pedagogical Psychology, Rivne State University of Humanities, Rivne, Ukraine; [6]Department of Theory and Methods of Physical Culture, Lviv State University of Physical Culture, Lviv, Ukraine and [7]Regional Alcohol and Drug Abuse Research (RADAR) Center, Ben-Gurion University of the Negev, Be'er Sheva, Israel

## Abstract

This study aims to discern similarities and differences associated with the impact of war on Ukrainian and Israeli women. We hypothesize that country affiliation significantly determines their mental health and psycho-emotional well-being. A total of 1,071 Ukrainian ($N = 601$) and Israeli ($N = 470$) women were surveyed online from September to December 2022 in Ukraine and November 2023 to March 2024 in Israel. Valid and reliable survey instruments were used to gather data about the fear of war, depression, loneliness, suicidal ideation and substance misuse. Fear of war and depression were higher among Israeli respondents. However, Ukrainian women reported more loneliness, substance use and psycho-emotional deterioration. Respondents from both countries did not show a different level of suicidal ideation. Two-way analysis of variance (ANOVA) results show fear of war associated with country and depression; and depression linked to country and increased alcohol use, especially among Ukrainian respondents. Comparative results partially confirm the study hypothesis. The impact of the war on Ukrainian and Israeli women has similar effects; however, differences exist that may be attributed to culture and adaptation to war length. Further research, including uniform data collection and analysis, is needed to determine the impact of war on women as well as their familial and work-related responsibilities that tend to increase during such conditions.

## Impact statement

The impact of war and violent conflict on the mental health and well-being of women remains understudied. The Russia–Ukraine and Arab–Israel wars take place in different social, cultural, psychological and economic conditions that impact women, their mental health and well-being. Despite ongoing prevention and coping measures associated with war and terrorism, Israeli women report a higher fear of war, depression and loneliness than from the Ukraine. This finding tends to be the result of data collected during an existential period for Israel and its people confronted with the confluence of factors including attack from multiple countries, international condemnation on many levels and internal political dissonance. The impact of war on Ukrainian and Israeli women is complex, and the similarities and differences identified may be related to culture, perception of war in society, its duration, intensity, internal and external international support and other factors. This study provides useful information about the impact of war on women that has relevance for policy and intervention services.

## Introduction

War, terrorism and violence with conditions of death and injury, refuge and relocation, food insecurity, draught, disease and vital service disruption are more prevalent now than at any time since the end of World War II (WHO, 2023; International Crisis Group, 2024a, 2024b). This situation has become more difficult because of the Russia–Ukraine and Arab–Israel wars with possible escalation beyond territorial or regional boundaries.

## Ukraine

On February 24, 2022, Russia invaded Ukraine provoking the most serious military conflict in Central Europe since 1945 and international concern about food production (Leal Filho et al., 2023). For the first time in its recent history, Ukraine is faced with armed conflict on its land accompanied by formidable challenges to address health and social service needs (Vus and Esterlis, 2022; Sokan-Adeaga et al., 2023; Kang et al., 2024). After 18 months, the Office of the United Nations High Commissioner for Human Rights (OHCHR) estimated 37,922 civilian casualties – 11,979 killed and 25,943 injured (OHCHR, 2024). According to the UN Refugee Agency, there are nearly 4 million internally displaced people in Ukraine and 6.8 million refugees mostly in Germany, Poland and Russia as of November 2024 (UN Refugee Agency, 2024). In Ukraine, national resilience was initially high due to a surge in unity and international support. However, over time, it has decreased due to the prolonged conflict, war fatigue and declining trust in the government (Reznik, 2023; Iancu, 2024).

## Israel

Since inception as a modern nation in 1948, Israel has been surrounded by hostile forces bent on its destruction. However, over time, its people have demonstrated preparedness for life under conditions of war and terror attack with resilience, social cohesion and service support (Iancu, 2024). Except for those with exemption to pursue full-time religious study, most Israeli men and women have military or national service experience contributing to their ability to cope and/or respond to traumatic events (Bleich et al., 2003; Ben-Tzur et al., 2021). On October 7, 2023, the Islamic Resistance Movement of "Hamas," a Gaza-based political organization, attacked rural settlements (i.e., kibbutzim and moshavim), development towns and a "Nova" music festival in the western region of country, resulting in the death of more than 1,100 civilians including infants, children, women and elderly people, as well as foreign agricultural workers. Among the physically injured and sexually assaulted, hundreds were kidnapped dead or alive and taken to Gaza (Gettleman et al., 2023; Williamson, 2023). Also, within the first 4 h of the attack, thousands of missiles were fired from Arab countries to Israel causing death, destruction, fear, stress and population displacement (Cortellessa, 2023; Impelli, 2023; Goldbart, 2024; Saidel et al., 2024; Statista, 2024).

## War and the health of women

Considerable research exists on military, political, economic, social and cultural factors associated with the present Russia–Ukraine and Arab–Israel conflicts (Tuşa, 2023; Finaud, 2024; Hassan and Mustafa, 2024; Marolov, 2024; Oleinik, 2024; Tzika, 2024). However, there is scant comparative information about the impact of war on the mental health and well-being of women from these countries. This may be attributed to the complex array of risk factors such as violent deaths, nonfatal physical injuries and disabilities resulting from mines and unexploded ordnance, sexual violence and unintended pregnancies (Jina and Thomas, 2013; Ajayi and Ezegbe, 2020; Shalak Markson and Nepal, 2023) as well as mental disorders including persistent sadness or feeling of hopelessness, eating and sleeping problems, substance misuse, isolation and suicidality or suicide attempts. Also, the lack of uniform data collection and analysis affects the amount of usable information available for informed decision-making and service intervention purposes on national and cross-national levels (Deb and Baudais, 2022; Dina Diatta and Berchtold, 2023; Jungblut, 2023; Krelinova et al., 2023). Regardless of such limitations, study findings from the Ukraine and Israel show women more than men with increased fear, depression, loneliness, suicidal ideation, post-traumatic stress disorder (PTSD) and substance misuse (Bendavid et al., 2021; Sheather, 2022; Kurapov et al., 2023a, 2023b; Pavlenko et al., 2023, 2024a, 2024b; Dopelt and Houminer-Klepar, 2024; Feingold et al., 2024; Groweiss et al., 2024; Hasson-Ohayon and Horesh, 2024; Katsoty et al., 2024; Levi-Belz et al., 2024; Palgi et al., 2024).

This study, based on uniform data collection and analysis, describes the demographic characteristics of Ukrainian and Israeli female survey respondents and reports the impact of war on their mental health and psycho-emotional well-being. We hypothesize that country affiliation is significantly associated with fear of war, depression, loneliness, suicidal ideation and substance misuse.

## Method

The Qualtrics software platform was used for this online survey that includes three data collection scales. The first was the 10-item Fear of War Scale (FWS) (Kurapov et al., 2023b). The agreement levels for the statements used are evaluated by a five-point Likert scale ranging from 1 (strongly disagree) to 5 (strongly agree). Higher total scores correspond with more fear of war. Exploratory factor analysis found the instrument as a two-factor model – both subscales with five items. The first subscale describes psychophysiological reactions to fear (e.g., "My heart is beating faster when I think about the war" and "I have a sleep disorder because I worry the war will get to me"). The second subscale is associated with existential fear reactions (e.g., "I am scared because war costs human lives" and "I am afraid the war will drag on for a long time"). Other scales used for this study include the nine-item Patient Health Questionnaire (PHQ-9) for measuring the severity of depression (Kroenke et al., 2001). Item 9 of the PHQ-9 ("Thoughts that you would be better off dead or hurting yourself in some way") was used as a dichotomous (yes/no) indicator of suicidal ideation. Also, the De Jong Gierveld six-item Loneliness Scale was used for the survey (Gierveld and Tilburg, 2006).

All instruments were translated from English to Ukrainian and Hebrew and back-translated. Cronbach's alpha scores for the scales used, totaled for both languages, are FWS = 0.862/0.805/0.852, PHQ-9 = 0.885/0.830/0.873 and Loneliness Scale = 0.794/0.763/0.764. Survey respondents provided information about their age, marital status, religiosity (secular/non-secular), substance use (i.e., tobacco, alcohol, pain relievers and sedatives) and psycho-emotional well-being. Statistical analyses were conducted using SPSS, version 29. Stepwise multiple regression, Pearson's chi-squared test for dichotomous variables, the Mann–Whitney test, *t*-test, one- and two-way analysis of variance (ANOVA), effect size measure, and 95% CI for mean were used for the data analysis.

The study was approved by the Ben-Gurion University institutional review board (approval: 22122022). For Ukrainian participants, ethical review and approval were waived for this study due to the anonymous data collection and reporting procedures used. The survey was conducted online and anonymously using the Qualtrics platform. Informed consent was contained in the introductory part of the questionnaire. In case of refusal to participate in the survey (i.e., negative response to the invitation to participate), an automatic exit from the online survey system occurred with the inability

to continue working. The start of the survey means respondent informed consent.

## Participants

A snowball, non-probability sampling technique was used for online data collection. The study cohort included 1,071 Ukrainian (56.1%) and Israeli (43.9%) women, 47.4% secular and 65.9% married/partnered, and most Israeli respondents reported military (86.6%) or national (13.2%) service.

The Ukrainian respondents were from five locations (i.e., Kyiv, Kharkov, Lviv, Chernigov and Rivne); the Israeli participants were from all regions of the country including those from the southern region of the country (36.8%), from the densely populated center with Jerusalem and Tel Aviv (42.2%), and from the north (21.0%). Data were collected from September to December 2022 in Ukraine and November 2023 to March 2024 in Israel. Table 1 provides demographic information about the participants.

**Table 1.** Demographic data

| | Total (n = 1,071) | Country | | |
| | | Ukraine (n = 601) | Israel (n = 470) | p-value[1] |
|---|---|---|---|---|
| Age, mean (SD) | 38.3 (13.5) | 39.1 (12.3) | 37.2 (15.0) | 0.022 |
| Median | 37.0 | 38.0 | 31.0 | |
| Range | (17–79) | (17–76) | (18–79) | |
| Religiosity, *n* (%) | | | | |
| Secular | 484 (47.4) | 118 (21.1) | 366 (79.4) | <0.001 |
| Non-secular | 537 (52.6) | 442 (78.9) | 96 (20.6) | |
| Marital status, *n* (%) | | | | |
| Married/partner | 672 (65.9) | 405 (71.9) | 267 (58.6) | <0.001 |
| Other | 347 (34.1) | 158 (28.1) | 189 (41.4) | |

[1]*p*-value of t-test and Chi-square test.

## Results

### Fear of war

For the two survey samples, the mean value of the FWS was 35.2 (SD = 7.5; 95%CI: 34.8–35.7), with a range of 10 to 50. Mean fear of war values were higher for Israeli than Ukrainian women ($t(957) = 8.062$; $p < 0.001$; $d = 0.526$). Two-way ANOVA did not show a significant difference in fear scores associated with country, religiosity and marital statuses. Regarding existential and psycho-physiological fear, Israeli respondents had higher levels of both types ($t(959) = 11.336$; $p < 0.001$; $d = 0.738$, and $t(958) = 3.801$; $p < 0.001$; $d = 0.248$, respectively).

### Depression

The PHQ-9 (i.e., depression severity) mean value was 10.4 (SD = 6.1; 95%CI: 10.0–10.8) with a range of 0–27. Mean depression values were higher among Israeli respondents ($t(914) = 9.795$; $p < 0.001$; $d = 0.653$). Two-way ANOVA did not evidence a significant difference in depression scores based on country, religiosity and marital/partner statuses. For interpretation (Kroenke et al., 2001), PHQ-9 scores were divided into five groups: 0–4 (no/minimal depression), 5–9 (mild depression), 10–14 (moderate depression), 15–19 (moderately severe depression) and 20–27 (severe depression). Figure 1 provides information on depression levels by country.

Regardless of country, Mann–Whitney test showed secular ($U = 67,634.0$; $Z = -7.813$; $p < 0.001$) and non-married/non-partnered ($U = 77,204.5$; $Z = -2.927$; $p = 0.003$) respondents with higher depression levels. Two-way ANOVA shows significantly different amounts of fear of war associated with country and depression ($F(4,857) = 13.664$; $p < 0.001$; partial $\eta^2 = 0.060$) (Figure 2).

Ukrainian and Israeli women did not show a different level of suicidal ideation (17.3% vs. 19.2%; $p = 0.463$). However, regardless of country, less suicidal ideation was found among those religious and married/partnered: $\chi^2(1, N = 878) = 7.039$; $p = 0.008$; $\varphi = 0.090$, and $\chi^2(1, N = 879) = 4.896$; $p = 0.027$; $\varphi = 0.075$.

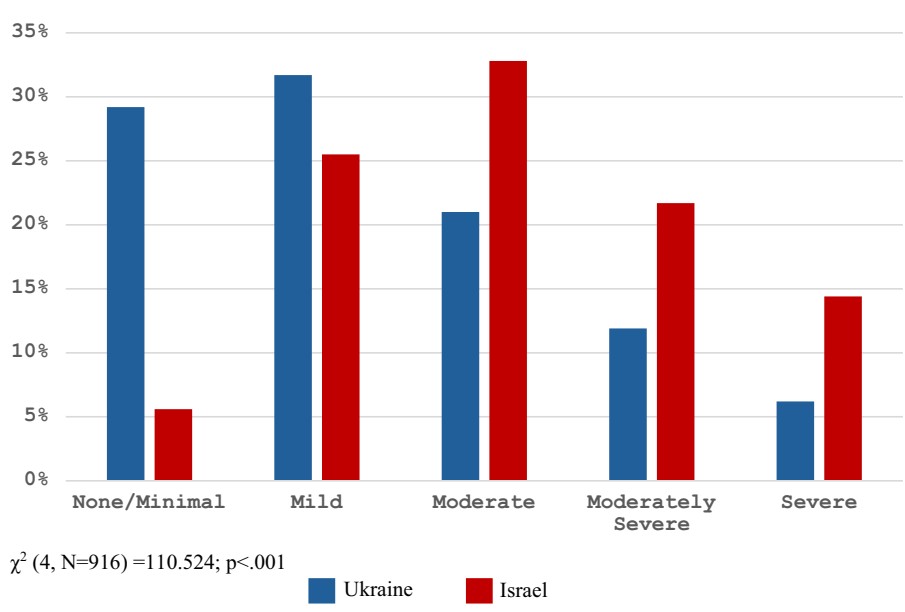

$\chi^2$ (4, N=916) =110.524; p<.001

**Figure 1.** Depression level by country.

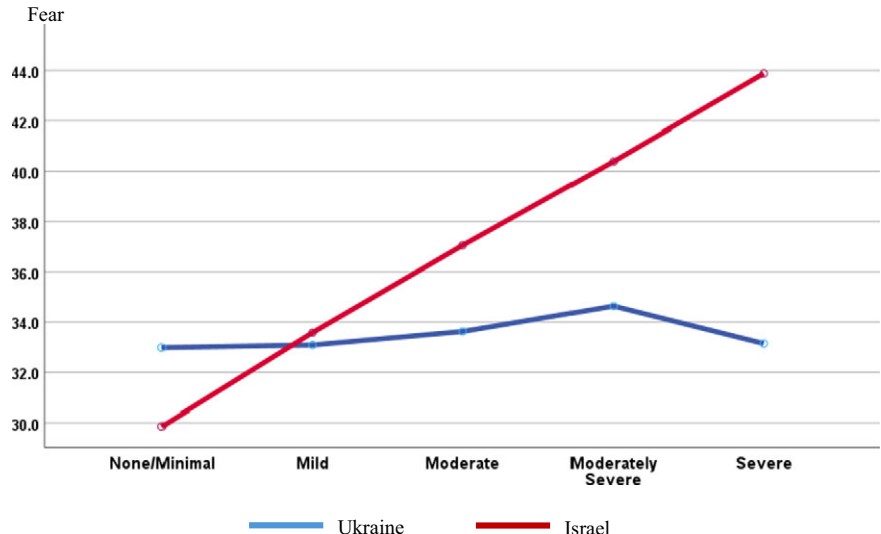

**Figure 2.** Fear of war by country and depression level.

### Loneliness

Israeli women reported a greater amount of emotional loneliness ($U = 80{,}591.0$; $Z = -4.895$; $p < 0.001$); and those from the Ukraine had more social and total loneliness (i.e., emotional and social loneliness combined) ($U = 70{,}547.5$; $Z = -7.787$; $p < 0.001$, and $U = 89{,}437.5$; $Z = -2.323$; $p = 0.020$, respectively). Regardless of the country, emotional loneliness was found more common among secular respondents ($U = 80{,}016.0$; $Z = -4.161$; $p < 0.001$) and social loneliness was more prevalent among those non-secular (i.e., religious) ($U = 82{,}676.0$; $Z = -3.479$; $p < 0.001$).

### Substance use

Last 30-day substance use increase of any type due to war was reported by 37.8% of the respondents. The rate of this behavior based on country was significantly different – 42.8% Ukrainian and 31.3% Israeli ($\chi^2(1, N = 909) = 12.503$; $p < .001$; $\varphi = 0.117$), especially for alcohol (18.4% versus 11.7%, $p = 0.006$; $\varphi = 0.090$), sedatives (21.8% versus 8.7%, $p < 0.001$; $\varphi = 0.176$) and binge drinking (10.1% versus 5.6%, $p = 0.015$; $\varphi = 0.081$). No significant differences were found for tobacco and pain relievers. Two-way ANOVA shows depression associated with country and increased alcohol use interaction ($F(1{,}838) = 22.784$; $p < 0.001$; partial $\eta^2 = 0.026$) (Figure 3). Regardless of country, respondents who reported increased alcohol use had higher scores of emotional, social and total loneliness: $U = 72{,}015.5$; $Z = -5.984$; $p < 0.001$; $U = 84{,}458.5$; $Z = -2.568$; $p = 0.010$; and $U = 75{,}025.5$; $Z = -4.896$; $p < 0.001$, respectively.

### Psycho-emotional state

Last month psycho-emotional deterioration was more prevalent among the Ukrainian (94.2%) than Israeli (80.1%) women

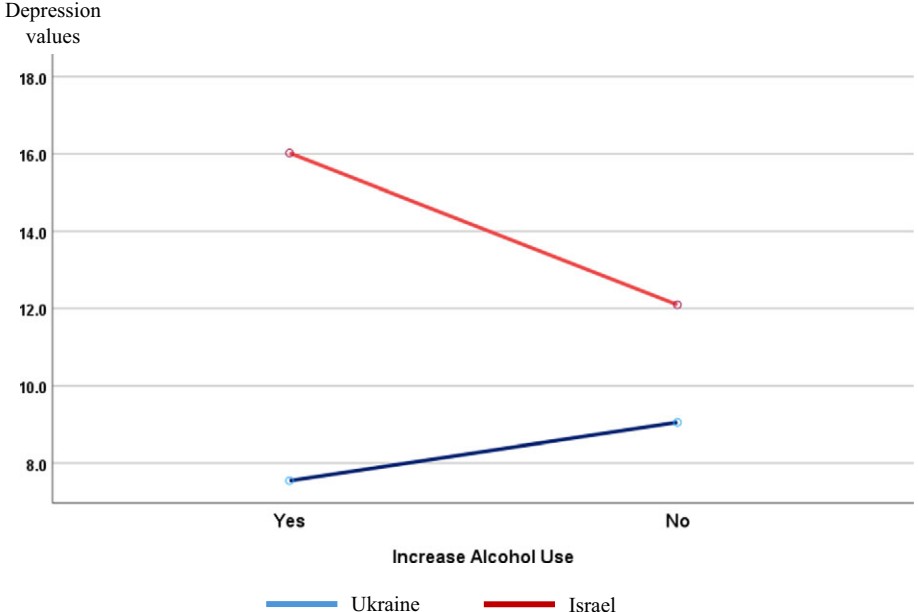

**Figure 3.** Depression by country and increase alcohol use.

$(\chi^2(1, N = 984) = 44.673; p < 0.001; \varphi = 0.213)$. Two-way ANOVA shows a significant difference in fear of war based on country and psycho-emotional deterioration $(F(1,921) = 13.865; p < 0.001;$ partial $\eta^2 = 0.015)$. Regardless of country, respondents who reported psycho-emotional deterioration tended to be lonelier $(U = 14,018.0; Z = -3.191; p = 0.001)$.

For all study respondents, stepwise regression analysis shows fear of war associated with country $(\beta = 0.301; p < 0.001)$, depression $(\beta = 0.236; p < 0.001)$, increased substance use $(\beta = -0.158; p < 0.001)$, psycho-emotional deterioration $(\beta = -0.144; p < 0.001)$, loneliness $(\beta = 0.208; p < 0.001)$, religiosity $(\beta = 0.117; p = 0.003)$ and marital status $(\beta = -0.083; p = 0.011)$. Additional independent variables (e.g., age and binge drinking) did not significantly increase the proportion of explained variance (i.e., adjusted $R^2 = 0.249$).

## Discussion and conclusion

A purposive sample of Ukrainian and Israeli women were compared during war conditions in terms of their fear of war, depression, loneliness, suicidal ideation, substance use and psycho-emotional well-being. Findings are consistent with other studies conducted on war in the Ukraine (Kurapov et al., 2023a, 2023b; Pavlenko et al., 2023, 2024a, 2024b), Israel (Kimhi et al., 2020; Solomon, 2020) and elsewhere (Carpiniello, 2023; Conflict Watchlist, 2024; M Ahmed et al., 2024; Our World in Data, 2024; The Peace Research Institute Oslo, 2024). Such results indicate that women, more than men, are at risk of acquiring acute and long-lasting health problems under such conditions (Arcel and Kastrup, 2004; Kastrup, 2006; Murthy and Lakshminarayana, 2006; Bogic et al., 2015).

When compared cross-nationally, findings partially confirm the study hypothesis. The impact of the war on Ukrainian and Israeli women shows many similar effects; however, differences exist that may be attributed to culture, adaptation to war length and conflict intensity. For example, Ukrainian women reported more substance use, binge drinking and psycho-emotional deterioration, but not depression associated with fear of war that may be an indication of adaptation to prevailing conditions that have lasted 3 years. Regardless of military or national service and preparedness for war, fear and depression levels were higher among Israeli women. This may be explained by the considerable impact of the October 7th invasion from Gaza, coordinated attacks from Lebanon, Syria, Iran, Iraq, Yemen and the West Bank; cyber insecurity; internal political dissonance; and widespread international condemnation that combined to be a disaster of major proportion for the country (Segell, 2025).

In severe life-threatening conditions, it is difficult to acquire timely and useful information (Institute of Medicine, 2015; Isralowitz, 2017). However, this study has relevance for informed decision-making associated with policy and services that may support the abilities of women to better address personal and familial responsibilities during war conditions as well as their lives going forward. Study findings evidence the impact the war has on women of Ukraine and those of Israel under attack from multiple countries, international condemnation on many levels and internal political dissonance. Shifting support for countries and people at war, evidenced by current political and government decision-making, gives substantive reason for the collection of relevant and useable information overtime and location.

## Limitations

This study has limitations. The study is based on convenience samples obtained using the snowball method without control of factors such as respondent economic and professional status, relations with family and friends, and country of origin status. The cross-sectional design and use of a purposive sample of women limit the ability to generalize study results. Furthermore, the use of an online survey made it possible for only those who had access to the Internet, and some potential survey participants were not able to participate due to communication and power supply failures.

**Open peer review.** To view the open peer review materials for this article, please visit http://doi.org/10.1017/gmh.2025.30.

**Data availability statement.** All data used in this publication are available upon request.

**Author contribution.** R.I. and A.R. conceptualized and designed the study. A.K., N.K., I.P. and SL.P. translated and adapted study instruments on Ukrainian and Hebrew. A.K., N.K., I.P., V.P., L.Z. and SL.P. conducted data collection and administering study. A.R. and SL.P. conducted the statistical analyses. R.I. and A.R. wrote the first draft of the manuscript. A.K., N.K., I.P., V.P., L.Z. and SL.P. reviewed the draft and provided critical feedback. All authors approved the final version of the manuscript.

**Financial support.** This research received no specific grant from any funding agency, commercial or not-for-profit sectors.

**Competing interests.** The authors declare that they have no known competing financial interest or personal relationships that could have appeared to influence the work reported in this study.

**Ethics statement.** The study was approved by the Ben-Gurion University institutional review board (approval: 22122022). Ethical review and approval were waived for this study by the Ethics Committees of the Lviv State University of Physical Culture, Ukraine; Faculty of Psychology and Natural Sciences of the Rivne State University of Humanities, Ukraine; Faculty of Psychology of the Taras Shevchenko National University of Kyiv; Institute of Psychology and Social Work; T.H. Shevchenko National University "Chernihiv Colehium," Ukraine; and Faculty of Psychology of the V. N. Karazin Kharkiv National University due to anonymous data collection and reporting procedures used.

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
