## [Reviewer Report]

The article is very interesting; however, there are some comments to improve it:

1. In the introduction, it is necessary to explain the cultural differences between the countries. For example, Israel is a country that has been through wars throughout its existence, surrounded by enemies on all borders and even within its borders. There is mandatory military service in the IDF, including for women, and Israel has financial stability, etc.

2. In the Methods section, it is necessary to add the analysis methods and the reasoning for their selection, how the online questionnaires were distributed, the age range of the participants, and whether women from both the north and south of Israel participated. Regarding Ukraine, there is a detailed explanation, but there is none for Israel.

3. Ethical aspects – It is very strange that the researchers did not require approval from an ethics committee in Israel given the anonymity of the questionnaires. After all, most studies are anonymous and still require approval. It would be advisable for the researchers to attach the letter from the ethics committee that approved this in an appendix.

4. Page 8, Line 16 in the Discussion – It is necessary to elaborate on the cultural and social differences that created the disparities in the results. This is precisely the essence of comparative research. In general, the researchers do not discuss all the findings in depth, and the Discussion section should be expanded.

5. The Conclusions section is missing, and there are no practical recommendations for women in conflict areas.

---

## [Reviewer Report]

General Comments

The manuscript presents an important and timely study on the mental health impact of war on Ukrainian and Israeli women. The topic is highly relevant, and the comparative approach is valuable. However, several key areas need improvement before publication.

1. Conceptual and Theoretical Issues

The paper would benefit from including more literature on the effects of war on mental health, especially in gendered contexts.

Cultural and contextual differences between Ukraine and Israel are mentioned but not deeply explored. Consider discussing how social, political, and historical factors shape mental health responses.

2. Statistical and Analytical Concerns

Please include effect sizes and confidence intervals to give a clearer picture of the findings.

Some key factors, like socioeconomic status, trauma history, military service (for Israeli respondents), and whether respondents are native-born or immigrants, are not controlled for. These could significantly impact results.

The use of Two-Way ANOVA is not well justified. Would a regression model or a more nuanced approach be more suitable?

3. Discussion and Organization

The discussion section feels a bit scattered. A clearer structure would help readers follow the key takeaways.

Some conclusions feel a little speculative (e.g., attributing differences mainly to culture). It would be great to back these up with more data or references.

The policy and intervention implications could be more developed. What practical recommendations can be made for mental health support in conflict settings?

4. Clarity and Formatting

The writing is clear overall, but some sections repeat information. Tightening the text will improve readability.

Final Recommendation: Major Revisions Needed.

This study has great potential, but it needs some key refinements to strengthen the methodology, analysis, and discussion. Addressing these points will make the manuscript even stronger and more impactful.